# Structures of H5N1 influenza polymerase with ANP32B reveal mechanisms of genome replication and host adaptation

Ecco Staller [1,6], Loïc Carrique [2,6], Olivia C. Swann[3], Haitian Fan [1,4], Jeremy R. Keown [2,5], Carol M. Sheppard [3], Wendy S. Barclay [3], Jonathan M. Grimes [2,7] ✉ & Ervin Fodor [1,7] ✉

Avian influenza A viruses (IAVs) pose a public health threat, as they are capable of triggering pandemics by crossing species barriers. Replication of avian IAVs in mammalian cells is hindered by species-specific variation in acidic nuclear phosphoprotein 32 (ANP32) proteins, which are essential for viral RNA genome replication. Adaptive mutations enable the IAV RNA polymerase (FluPolA) to surmount this barrier. Here, we present cryo-electron microscopy structures of monomeric and dimeric avian H5N1 FluPolA with human ANP32B. ANP32B interacts with the PA subunit of FluPolA in the monomeric form, at the site used for its docking onto the C-terminal domain of host RNA polymerase II during viral transcription. ANP32B acts as a chaperone, guiding FluPolA towards a ribonucleoprotein-associated FluPolA to form an asymmetric dimer —the replication platform for the viral genome. These findings offer insights into the molecular mechanisms governing IAV genome replication, while enhancing our understanding of the molecular processes underpinning mammalian adaptations in avian-origin FluPolA.

Influenza viruses have a negative-sense segmented RNA genome. The 5′ and 3′ termini of each viral RNA (vRNA) segment associate with FluPol, while the rest of the RNA is packaged in a scaffold of viral nucleoprotein (NP) to form a viral ribonucleoprotein (vRNP)[1–3]. FluPol, composed of the polymerase acidic (PA), polymerase basic 1 (PB1) and polymerase basic 2 (PB2) subunits (Supplementary Fig. 1a), transcribes viral genes and replicates the vRNA in the nucleus of infected cells. These activities require conformational rearrangements that are regulated by interactions with host and viral factors, including newly synthesised FluPol. Transcription is a primer-dependent process that requires capped RNA fragments derived from host RNAs. To access these, the FluPol of incoming vRNPs docks onto the C-terminal domain (CTD) of host RNA polymerase II (RNAP II)[4–8]. Replication is a primer-

independent two-step process: first, a complementary RNA (cRNA) is synthesised, which then acts as a template for additional vRNA synthesis. cRNA and vRNA are assembled into complementary ribonucleoproteins (cRNPs) and vRNPs, respectively, to avoid exposure of cRNA and vRNA to the nucleoplasm, where it may get degraded by nucleases or sensed by innate immune factors[9]. Genome replication is dependent on host acidic nuclear phosphoprotein 32 (ANP32) proteins[10,11] and FluPol oligomerisation[12,13].

Recently, we determined structures of influenza C virus polymerase (FluPolC) forming replication platforms with chicken and human ANP32A[14]. The replication platform comprises an asymmetric dimer of FluPolC molecules, one of which functions as the replicase (FluPol$_R$) and the other (FluPol$_E$) encapsidates the genomic RNA

[1]Sir William Dunn School of Pathology, University of Oxford, Oxford, UK. [2]Division of Structural Biology, Centre for Human Genetics, University of Oxford, Oxford, UK. [3]Section of Molecular Virology, Imperial College London, London, UK. [4]Present address: School of Basic Medical Sciences, Zhejiang University School of Medicine, Zhejiang University, Hangzhou, China. [5]Present address: School of Life Sciences, University of Warwick, Coventry, UK. [6]These authors contributed equally: Ecco Staller, Loïc Carrique. [7]These authors jointly supervised this work: Jonathan M. Grimes, Ervin Fodor. ✉e-mail: jonathan.grimes@strubi.ox.ac.uk; ervin.fodor@path.ox.ac.uk

product emerging from FluPol_R. The N-terminal leucine-rich repeat domain of ANP32A (ANP32A^LRR) bridges the two FluPolC molecules, while the unresolved C-terminal low-complexity acidic region (LCAR) is believed to recruit NP[15]. The second step of replication—vRNA synthesis from a cRNA template—may require alternative, symmetric FluPol dimers for template realignment[16]. Along with vRNPs, free Flu-Pol and NP, ANP32 proteins are essential for influenza A virus (IAV), influenza B virus and influenza C virus (ICV) replication, supporting vRNA as well as cRNA synthesis[10,14,17–19]. Although ANP32 proteins are highly conserved among vertebrates, genetic variation, such as a 33 amino acid insertion in avian ANP32A[17,20], or substitutions in amino acid residues 129, 130 or 156[10,11,21–23], can affect influenza virus replication. IAVs naturally reside in the gastrointestinal tract of aquatic birds like ducks and geese; avian FluPolA activity is stymied in mammalian cells due to mismatched ANP32 proteins. Therefore, to cross into mammals, IAVs undergo mutations in the genes encoding their FluPolA subunits. The best-known example of a mammalian adaptation is a glutamate-to-lysine substitution in PB2 (PB2 E627K)[24]. Despite an increasing understanding of avian IAV adaptation, the molecular mechanisms driving such mutations have remained elusive.

Here, we use cryo-EM to characterise the interaction of human ANP32B with FluPolA derived from the avian H5N1 strain A/turkey/Turkey/1/2005 (Tky05), which naturally carries the mammalian adaptations PB2 627K and PA 383D[25]. Tky05 FluPolA acquired additional mammalian adaptations during passaging experiments in human cells lacking ANP32A and ANP32B (dKO)[26]. These substitutions—PB1 K577E and PA Q556R—allowed Tky05 FluPolA to co-opt an alternative host factor, ANP32E, to support its replication. Importantly, this adapted polymerase can still be supported by ANP32B[26]. Substitutions of the lysines at positions 577 and 578 of the PB1 subunit are common in avian-origin FluPolA adapting to replication in mice or mammalian cells; loss of either basic residue leads to enhanced virulence in mice and increased FluPolA activity, as well as reduced formation of symmetric dimers[27–30]. PA Q556R has also been widely described as a mammalian adaptation[30–35]. Furthermore, it evolved in an H9N2 virus infecting *Anp32A* gene-edited chickens[36], and has been shown to enhance ANP32B binding to FluPolA[26]. We reasoned that disruption of symmetric dimer formation due to PB1 K577E, in combination with enhanced interaction with ANP32B, would be useful in obtaining a stable FluPolA–ANP32B replication complex. All experiments presented in this study were carried out with H5N1 Tky05 FluPolA with the PB1 K577E and PA Q556R amino acid changes. We uncover complexes of ANP32B with monomeric and dimeric FluPolA, revealing the structure of an IAV replication platform. In agreement with a recent publication[26], we propose that some mammalian adaptations seen in avian-origin FluPolA arise to allow assembly of replication platforms with mammalian ANP32 proteins.

## Results

### Human ANP32B forms a complex with monomeric H5N1 FluPolA
To gain insight into the structural basis of genome replication in IAV, we employed a recombinant baculovirus co-expressing Tky05 FluPolA subunits PB1, PB2 and PA, along with human ANP32B, in Sf9 insect cells. We then purified the FluPolA–ANP32B complexes via affinity and size exclusion chromatography (SEC). Cryo-EM analysis revealed that FluPolA is present in equilibrium between monomeric and dimeric forms, mostly bound to ANP32B (Supplementary Fig. 2). We did not observe symmetric dimers or higher-order oligomers in the samples. We determined the structure of a monomeric FluPolA in complex with ANP32B at a final resolution of 3.1 Å (Fig. 1a, Supplementary Figs. 2 and 3 and Supplementary Table 1). The endonuclease domain of PA (PA^Endo), the CTDs of PB2 (PB2-C) and most of the highly acidic ANP32B^LCAR (amino acids 160–251) remained flexible and were not resolved. The ANP32B^LRR domain (amino acids 1–149) makes extensive contacts with the large CTD of PA (PA-C). Amino acid residues N129

and D130, which are essential for interaction between FluPol and ANP32A proteins[11,37], form hydrogen bonds with PA K635. N129 also interacts with the first methionine of the PB1 subunit (Fig. 1b). The PA 550 loop (residues 550–560)[38] forms a β-hairpin that inserts between the ANP32B^LRR and the resolved N-terminal section of the ANP32B^LCAR (residues 150–159), with the hydrophobic residues A553 and V554 contacting the concave face of the ANP32B^LRR. PA R551 forms a salt bridge with ANP32B D159, while PA R559 interacts with ANP32B D119 and D146, in addition to π-stacking with F121 (Fig. 1c). PA R556 forms two salt bridges, with ANP32B E154 and D157, confirming its role in strengthening the interaction with ANP32B.

The PA-C domain of FluPolA not only binds ANP32B, it also interacts with the host RNAP II CTD[5,8] during viral transcription[5,8,9,39]. Comparison of the ANP32B and RNAP II CTD binding sites suggests that their binding is mutually exclusive (Fig. 1d). We performed competition experiments by immobilising RNAP II CTD peptides[6] on streptavidin resin, followed by incubation with FluPolA in the presence or absence of ANP32B. As expected, FluPolA binds the serine 5 phosphorylated (S5P) version of the CTD, a hallmark of RNAP II transcription initiation[40], but in the presence of ANP32B binding is reduced >6-fold (Fig. 1e). FluPolA pull-down is dependent on ANP32B concentration, while absence of ANP32B signal on the gel confirms that it does not bind CTD peptides itself, i.e. CTD binding is mediated solely by FluPolA (Supplementary Fig. 4). These data suggest that ANP32B can outcompete RNAP II for binding to FluPolA, in a concentration-dependent manner. Altogether, our data suggest that newly synthesised, RNA-free FluPolA binds ANP32B as it enters the cell nucleus, and that this interaction prevents binding to the RNAP II CTD.

### Dimeric H5N1 FluPolA forms a replication platform with human ANP32B
We next determined the structure of the Tky05 replication platform, comprising an asymmetric dimer of FluPolA and ANP32B, to a final resolution of 3.2 Å (Fig. 2a, Supplementary Figs. 2 and 3, Supplementary Table 1 and Supplementary Movie 1). In this complex, the active site of the FluPol_R PA^Endo domain is buried and thus cannot perform cleavage of capped host RNA, i.e. FluPol_R is transcription incompetent. Its orientation is stabilised by the arrangement of the PB2-C domains: the PB2 mid-link (PB2^Mid-link) and cap binding domain (PB2^CBD) pack against the PB1 palm subdomain (PB1^Palm) to form the product exit channel (Supplementary Fig. 1c). The PB2^627 domain sits at the dimer interface, while the PB2 nuclear localisation signal (PB2^NLS) domain packs against the PB2 lid (PB2^Lid) and PA^Endo domains, locking the overall FluPol_R conformation (Fig. 2b and Supplementary Fig. 1b).

The PB2-C domains of FluPol_E are arranged differently from published transcriptase (FluPol_T) or replicase (FluPol_R) conformations[14,16,41]. In those previous structures, the PB2^627 and PB2^NLS domains pack against the PA-C domain (Supplementary Fig. 1b and Supplementary Movie 1), while in this study the PB2^Mid-link and PB2^CBD of FluPol_E pack against the PB2 N2 (PB2^N2) domain (Fig. 2b and Supplementary Fig. 1b). In the transcriptase and replicase conformations, the PB2^Lid domain packs against the PB1 thumb subdomain (PB1^Thumb) and the C-terminal region to form the template exit channel (Supplementary Fig. 1b, c). In contrast, in FluPol_E the PB2^Lid flips over to mediate the stacking between PA^Endo and PB2^CBD via its C-terminal α-helix (Fig. 2b and Supplementary Fig. 1b, d). This position of the PB2^Lid and PA^Endo domains is unique—to the best of our knowledge a direct interaction between the PB2^CBD and the PA^Endo or PB2^Lid domains has not been observed before. To assume this configuration, PA^Endo rotates by 70° so that its loop^51–72 stacks against the PB2^CBD domain, sandwiching the PB2^Lid between the PA^Endo α-helix^84–98 and the PB2^CBD domains (Supplementary Fig. 1d). The interface is stabilised through extensive hydrophobic and hydrogen bonding interactions (Fig. 2c) and secures the overall FluPol_E conformation. This explains why the PA loop^51–72 was found to be essential for viral replication but not transcription[42]. Without the loop the

FluPol_E configuration cannot form, and without FluPol_E the replication platform cannot form.

The interaction between FluPol_R and FluPol_E is predominantly mediated by their respective PB2 and PA subunits, forming a large interface. The PA-C domain and PB1 β-hairpin of FluPol_E stack against the PB2^N2 and PB2^Mid-link domains of FluPol_R (Fig. 2d, e), while the PB2^627 domain of FluPol_R interacts with the PA-C and the PB2^627-NLS domains of FluPol_E (Fig. 2f). The conformation of ANP32B in the replication platform is practically identical to that in the ANP32B-bound monomeric

FluPolA structure (root-mean-square deviation = 1.056 Å). Both structures have clear density for the ANP32B^LRR (amino acid 1–149) and the N-terminal region of the ANP32B^LCAR (residue 150–159). However, in the replication platform, ANP32B interacts with the FluPol_E PB2^627 domain, while we observe no direct contacts with FluPol_R. Of note, the canonical mammalian adaptive residue PB2 K627 of FluPol_E forms a salt bridge with ANP32B E151 (Fig. 2g). PB2 R630 forms a hydrogen bond with the main chain of ANP32B E154, as well as π-stacking with ANP32B P156 (Fig. 2g). A proline-to-serine substitution at position 156 of swine

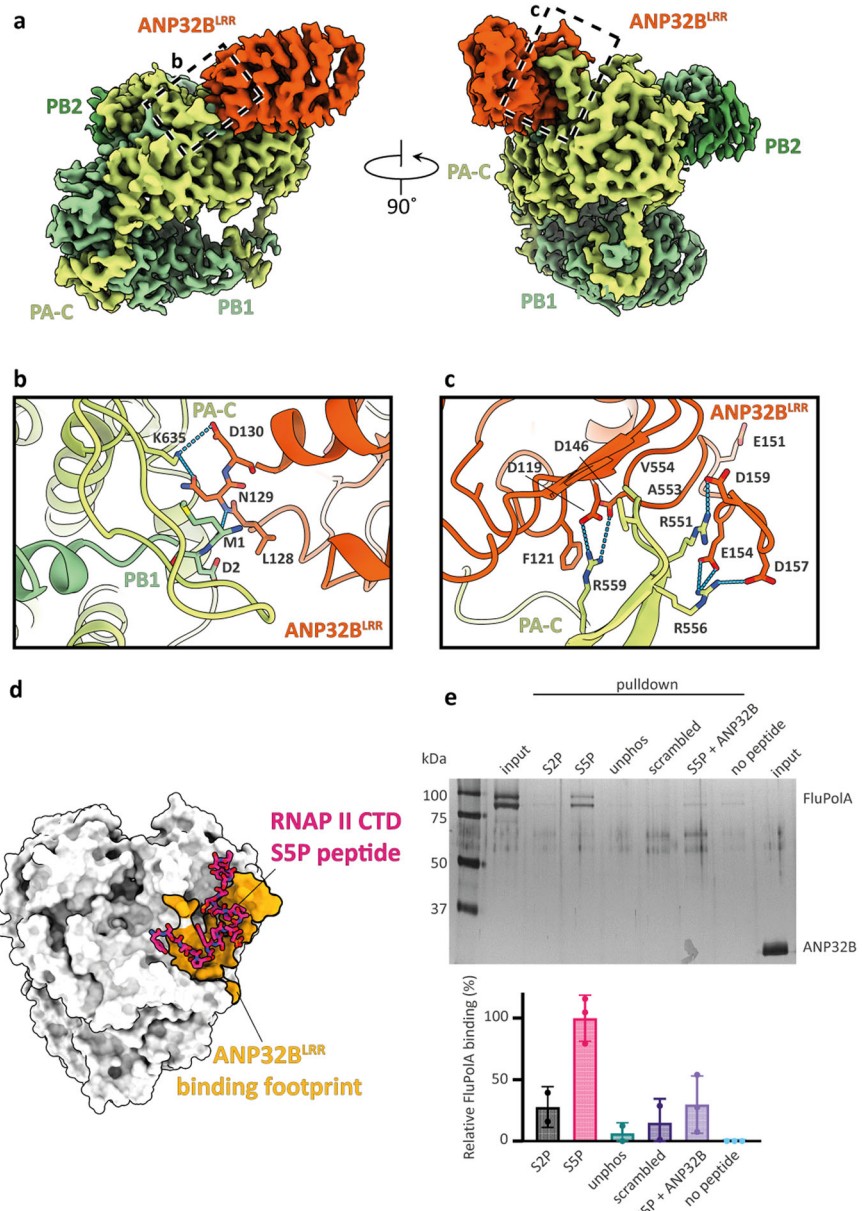

**Fig. 1 | Human ANP32B forms a complex with monomeric H5N1 FluPolA. a** Cryo-EM map of FluPolA heterotrimer (green) and ANP32B^LRR (orange). **b**, **c** Close-up views of the ANP32B^LRR–PA interactions. Dashed lines indicate hydrogen bonds. The dashed rectangles in panels (**a**, **b**) denote the close-up view positions in (**b**, **c**). **d** Surface representation of FluPolA with the binding footprint of ANP32B^LRR highlighted in orange (this study) and the RNAP II CTD serine 5 phosphorylated (S5P) peptide (PDB 6FHH). **e** Pull-down experiment showing Tky05 FluPolA (PB1 577E and PA 556R) binding to RNAPII S5P CTD peptide (lane 4), but not in the presence of ANP32B (lane 7). Controls include RNAP II serine 2-phosphorylated (S2P) CTD peptide (S2P), unphosphorylated peptide (unphos), scrambled peptide and no peptide. Relative FluPolA binding levels to the peptide (bottom) are

shown as a percentage of S5P binding (set at 100%). Band intensity was quantitated using Image J and analysed in GraphPad Prism 10. The signal in the no peptide lane (lane 8) was set to zero and subtracted from the other lanes, essentially rendering background signal (i.e. FluPolA binding to sepharose beads in the absence of CTD peptide) zero. Data represent mean values ± standard deviation (*n* = 3 biologically independent experiments). Ordinary one-way ANOVA with Dunnett's multiple comparison test was used to assess significance, with the confidence interval set at 95%. *P* values are as follows: S5P vs S2P: *P* = 0.0043; S5P vs unphos: *P* = 0.0007; S5P vs scrambled: *P* = 0.0014; S5P vs S5P + ANP32B: *P* = 0.0025; and S5P vs no peptide: *P* = 0.0002. Source data are provided as a Source data file.

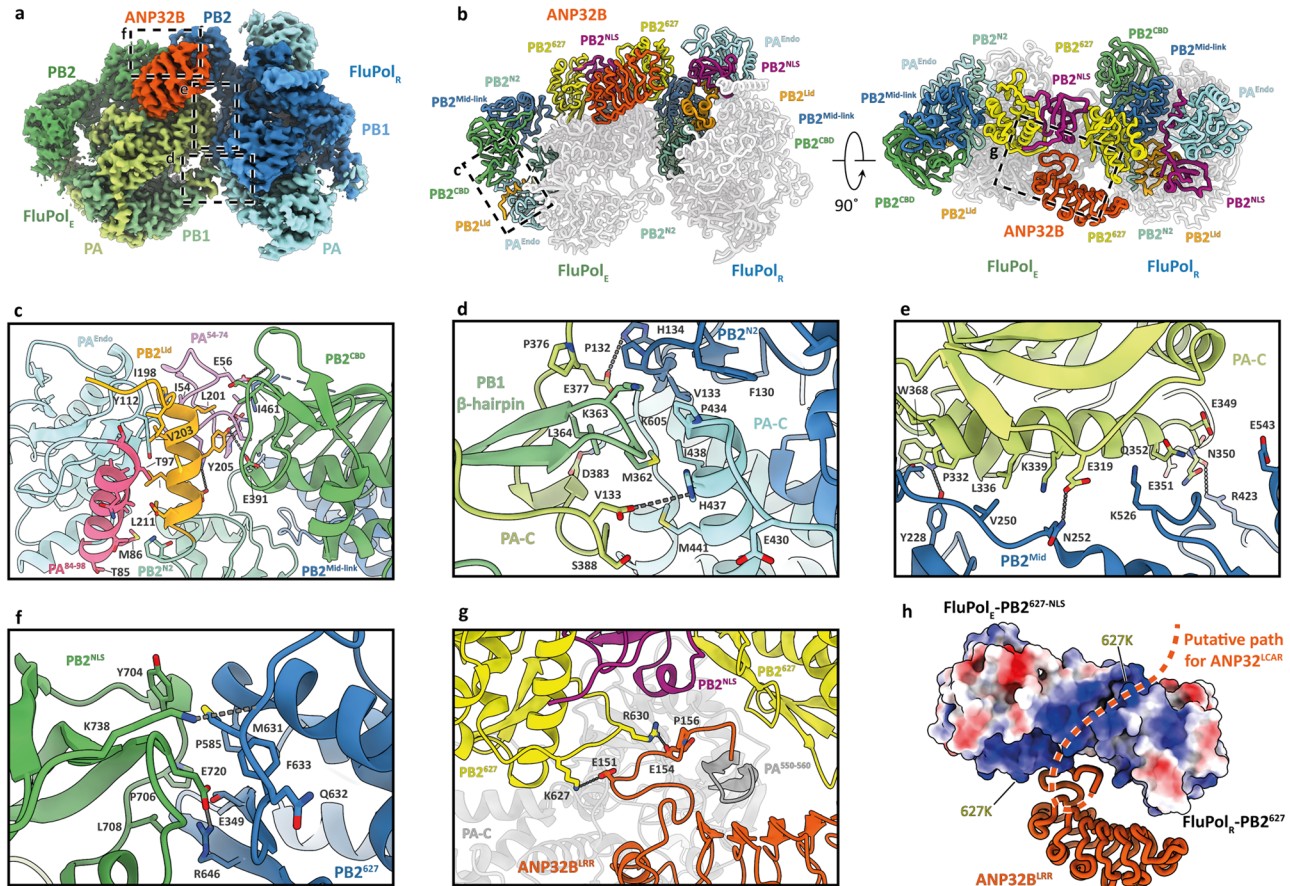

**Fig. 2 | H5N1 FluPolA dimer forms a replication platform with human ANP32B.**
**a** Cryo-EM map of IAV replication platform composed of a replicase heterotrimer (FluPol$_R$, blue), an encapsidating heterotrimer (FluPol$_E$, green) and ANP32B$^{LRR}$ (orange). **b** Model showing the organisation of the PB2 flexible domains and PA$^{Endo}$ in the replication platform. **c** Close-up view of the PA$^{Endo}$, PB2$^{Lid}$ and PB2$^{CBD}$ interface. **d**–**f** Close-up views of the FluPol$_R$-FluPol$_E$ interface. **g** Close-up view of the PB2$^{627}$-ANP32B interface. Dashed lines indicate hydrogen bonds. For panel **c**–**g**, the dashed rectangles in panels **a** and **b** denote the close-up view position. **h** Electrostatic iso-surface of the FluPol$_E$ PB2$^{627-NLS}$–FluPol$_R$ PB2$^{627}$ domains showing a positively charged groove adjacent to ANP32B$^{LRR}$, with the dashed line highlighting the putative path of the ANP32$^{LCAR}$ domain. The orientation is identical to the rotated view in panel (**b**).

ANP32A allowed avian FluPolA to replicate to some extent in porcine cells[23,43], lending credence to the idea of pigs as intermediary hosts for avian IAVs. Our structure suggests that an ANP32 P156S mutation would allow additional hydrogen bonding with FluPol$_E$. Mutagenesis of PB2 residues 627 and 630, and ANP32B residues 151 and 156 leads to significantly reduced FluPolA activity in vRNP reconstitution assays, suggesting this interface is biologically relevant (Supplementary Fig. 5).

PB2 K627 of FluPol$_R$ is situated in a positively charged groove that may be a binding site for part of the unresolved section of the ANP32B LCAR domain (residues 160−251) (Fig. 2h). The ubiquitous PB2 627E-to-K switch, observed when avian IAVs spill over into mammals, likely allows a region of the fully acidic mammalian LCAR domain (for example, residues 176-DEEDEDD−183 in human ANP32B; acidic residues underlined) to interact with FluPolA, as suggested by Carrique et al.[14]. Avian ANP32A contains acidic as well as basic residues in this region (176-VLSLVKDR−183; acidic and basic residues underlined), allowing interaction with either glutamate or lysine[14] (Supplementary Fig. 6).

**Comparison of the IAV and ICV replication platforms**
The marked similarity of the ANP32A and ANP32B LRR domains[44] allows direct comparison between the IAV (with ANP32B) and ICV (with ANP32A) replication platforms, despite the fact that IAVs and ICVs, although closely related, represent different influenza types that diverged around 8,000 years ago[45]. FluPolA and FluPolC are only about 20−40% identical at the amino acid level[46], so differences in their

interactions with ANP32 proteins of their respective hosts (humans and swine only in the case of ICV; a wide variety of birds and mammals for IAV) are to be expected. Nevertheless, the replication platforms share a similar overall arrangement, albeit exhibiting some striking differences (Supplementary Movie 1).

Notably, in the IAV replication platform, only the C-terminal region of the ANP32B$^{LRR}$ binds to the complex, exclusively interacting with FluPol$_E$ (Fig. 3a). This reinforces the essential role of ANP32 residues N129 and D130 in supporting replication. Unlike the ICV complex, no direct interaction is observed between the N-terminal region of the ANP32B$^{LRR}$ and FluPol$_R$, suggesting that IAV has evolved to optimise the binding of ANP32 exclusively to FluPol$_E$. We speculate that a single, optimised binding interface may allow IAV to target a wide range of host animals.

Insertions into regions of the ICV P3-C domain (P3 being the ICV equivalent of PA), in particular the loops between residues 490 and 500 and the 550 loop (Fig. 3c), lead to ANP32B binding to the IAV FluPol$_E$ PA-C in an alternative orientation (Fig. 3b). In ICV, the P3 550 loop is positioned between the PB2$^{627-NLS}$ domain and the ANP32A$^{LCAR}$, preventing their interaction (Fig. 3d). In IAV, the shorter PA 550 loop is inserted between the ANP32B$^{LRR}$ and the ANP32B$^{LCAR}$ and, since the PA$^{490-500}$ loop is also shorter, the PB2$^{627-NLS}$ is rotated towards the ANP32B$^{LCAR}$, promoting direct interactions (Fig. 3d). These differences also affect how the remaining PB2-C domains are arranged: in IAV, the stacking of the PA$^{Endo}$ domain against the PB2$^{CBD}$ is mediated by PA loop$^{51-72}$, which is not conserved in ICV (Fig. 3c).

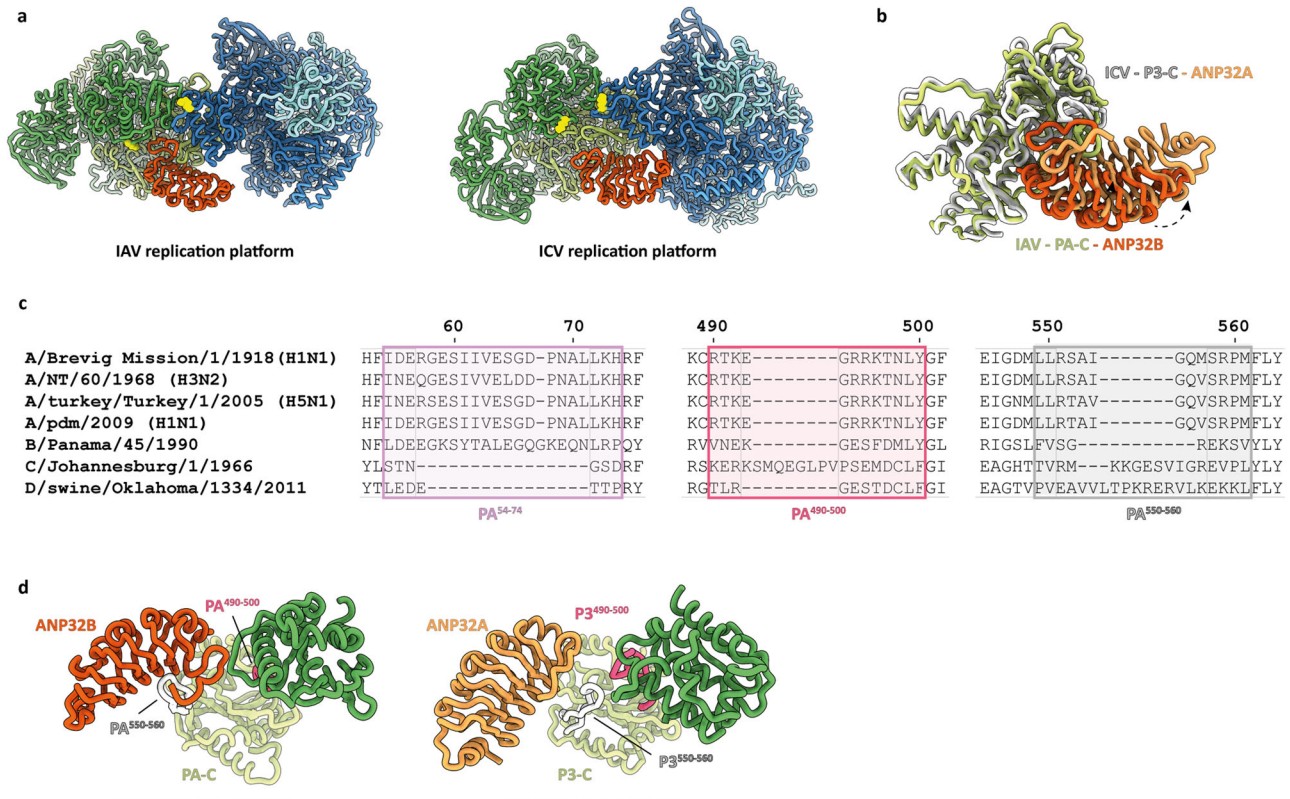

**Fig. 3 | Comparison between IAV and ICV replication platforms. a** Overall assembly of IAV and ICV replication platforms composed of FluPol$_R$ (blue), FluPol$_E$ (green) and ANP32$^{LRR}$ (orange; human ANP32B in the IAV platform; chicken ANP32A in the ICV platform). FluPolA PB2 K627 and its FluPolC equivalent K649 on both FluPol$_R$ and FluPol$_E$ are highlighted in yellow. **b** Comparison of the binding of human ANP32B$^{LRR}$ (dark orange) and chicken ANP32A$^{LRR}$ (light orange) to the PA-C domain of IAV (green) and equivalent P3-C of ICV (white). **c** Sequence alignment of the PA/P3 subunits of different influenza viruses. Influenza C virus (C/Johannesburg/1/1966) has elongated 490–500 and 550–560 loops, compared with influenza A virus, but it lacks a 51–72 loop. **d** Variation in the PA$^{490–500}$ and PA$^{550–560}$ loops between IAV and ICV affects the positioning of ANP32$^{LRR}$ (orange) and PB2$^{627}$ (dark green) relative to the PA/P3-C domain (light green). The ICV replication platform structure in panels (**a**, **b** and **d**) is based on PDB 6XZR.

In summary, like their interaction with the CTD of host RNAP II[39], differences in ANP32 protein binding between IAV and ICV likely reflect their evolutionary distance, and is informative with regard to key amino acid interactions.

**Molecular basis of mammalian adaptations in avian FluPolA**

Mutations in the viral polymerase (Fig. 4a) are vital to the process of avian IAV adaptation to mammalian hosts[47]. PB2 E627K is the hallmark of mammalian adaptation; all 20th century IAVs circulating in humans had this signature. In contrast, the 2009 pandemic H1N1 viruses had PB2 627E−their replication in mammals is attributed predominantly to a glutamine-to-arginine mutation at position 591 of PB2[48,49]. Our structure shows that K627 and Q591 are adjacent in FluPol$_E$ and could substitute each other to form a salt bridge with ANP32 E151 upon mutation into a basic residue (Fig. 4b). The more recently identified adaptive mutation PB2 M631L is in the same FluPol$_E$ cluster (Fig. 4b). In combination with PA E349K, this mutation allowed an avian H9N2 FluPolA to replicate using the normally non-functional chicken ANP32B and ANP32E (as opposed to the proviral chicken ANP32A), as well as human ANP32 proteins[36]. Other mammalian adaptations in the PB2 subunit of avian FluPolA−including 2009 pH1N1 viruses−are T271A and D701N[50,51]. It has been suggested that PB2 Q591R, E627K and D701N specifically adapt avian FluPolA to mammalian ANP32 proteins[52], while we hypothesise that T271A stabilises the FluPol$_E$ conformation through interaction with the 424 loop of PB2$^{CBD}$[53]. Of note, on FluPol$_R$ Q591, K627 and M631 also form a cluster (Fig. 4a), on the interface with the PB2$^{627}$ domain of FluPol$_E$, which is the presumed

location of the LCAR domain of ANP32. Thus some residues may fulfil a double function of strengthening FluPol$_E$ (via ANP32 residue 151), as well as FluPol$_R$ (through LCAR residues) binding to ANP32. Nevertheless, it is not entirely clear from the current structure how the 33 amino acid insertion in avian ANP32A allows both PB2 627E and 627K FluPolA activity.

Considering FluPolA adaptations that occur in nature when avian IAVs, including the highly pathogenic H5Nx and H7Nx subtypes, spill over into mammals (Fig. 4a and Supplementary Table 2), it appears that they fall in one of two broad categories. Substitutions like E627K in PB2 and Q556R in PA strengthen interactions between mammalian ANP32 proteins and FluPol$_E$. Adaptations including PA E349K and PB2 K526R may shift the FluPolA oligomeric equilibrium (in particular, asymmetric vs symmetric dimers) in favour of the asymmetric replication platform, thus promoting replication.

Besides the PB2 591/627/631 clusters, we observe a potential mutational hotspot at the PA-PA interface of the platform. Arai and colleagues[54] identify a pair of PA substitutions−S388R and A448E−from an Egyptian H5N1 strain that replicates to high titres in human cells. These residues sit on opposite sides of the FluPol$_E$−FluPol$_R$ interface and may enhance affinity between the molecules by forming an additional hydrogen bond (Fig. 4c). One of the mammalian adaptations that is already present in wildtype Tky05 FluPolA, PA N383D, maps a short distance across from K605 of the FluPol$_R$ PA subunit. Here a hydrogen bond can form between the FluPol$_E$ D383 and the FluPol$_R$ K605 (Fig. 4c). Clustered mutational hotspots are potential targets for antiviral intervention.

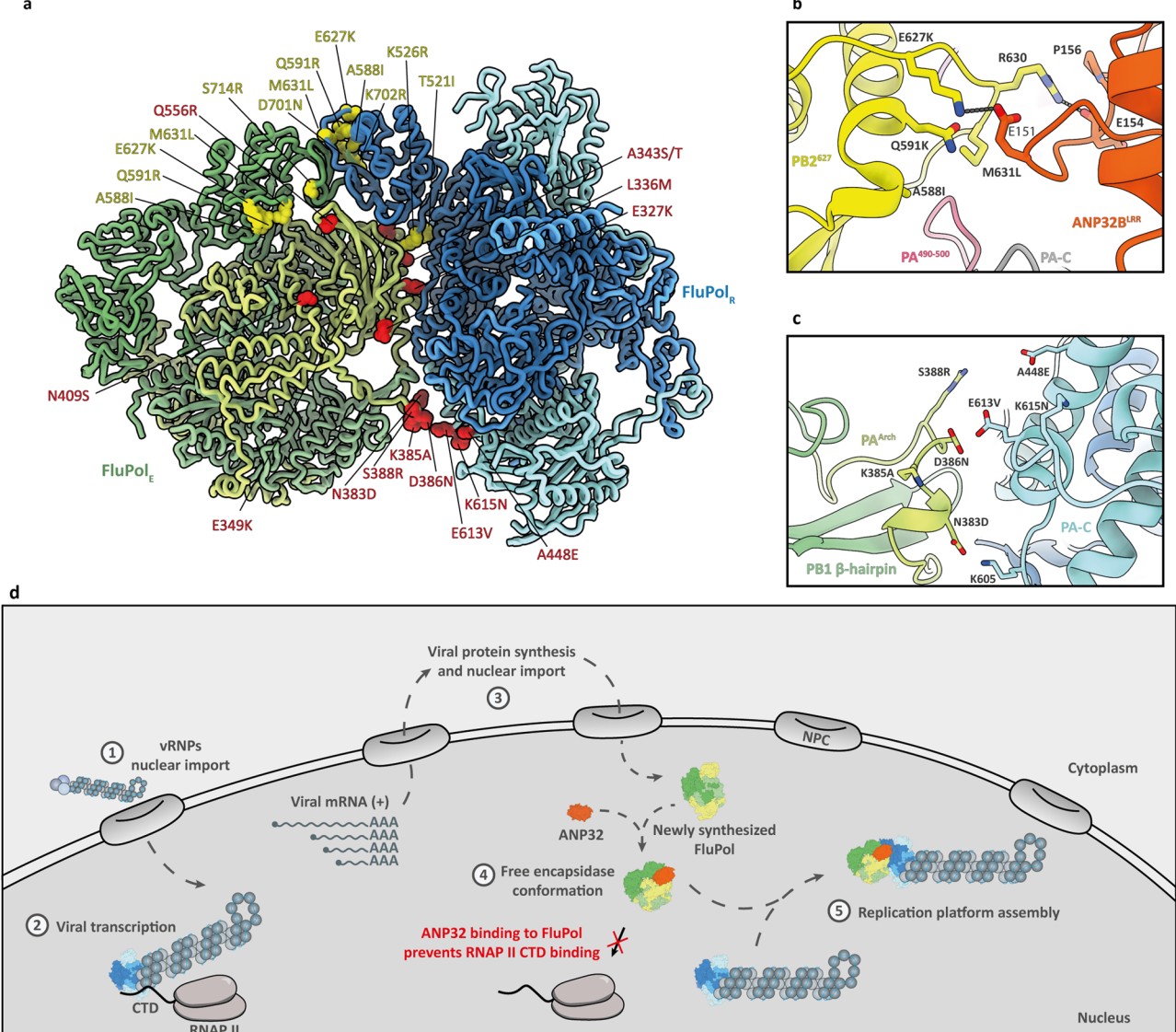

**Fig. 4 | Adaptive mutations on the IAV replication platform. a** Overall assembly of IAV replication platform (without ANP32B), mapping mammalian FluPolA adaptations potentially affecting either ANP32B binding or FluPol$_R$–FluPol$_E$ interactions. Mutations in PA are highlighted in red; PB2 adaptations in green. See also Supplementary Table 2. **b** Close-up view of adaptive cluster including PB2 residues 591, 627, 630 and 631, and their interactions with ANP32B residues 151, 154 and 156. **c** Close-up view of potential adaptive cluster on the FluPol$_E$–FluPol$_R$ PA–PA interface. Residue 383 of FluPol$_E$ already is aspartate in wildtype Tky05 FluPolA. Also shown is a pair of adaptations found in human-adapted H5N1 FluPolA, namely

S388R (FluPol$_E$) and A448E (FluPol$_R$). **d** Model for the role of ANP32 proteins in the assembly of the IAV replication platform. Upon nuclear entry (1), vRNP-associated FluPolA docks onto the CTD of RNAP II for primary transcription (2). Viral mRNAs are translated in the cytoplasm and newly synthesised FluPol enters the nucleus (3), where it associates with ANP32 and adopts an encapsidating conformation (4). The RNA-free FluPol$_E$–ANP32 complex then associates with vRNP-associated FluPol, stabilising it in a replicase conformation (FluPol$_R$) and forming a replication platform (5).

## Discussion

The insights gained in this study enable us to propose a model for the assembly of the IAV replication platform (Fig. 4d). Upon viral infection, the vRNPs are trafficked into the nucleus. Through the interaction between the host RNAP II CTD and the PA-C domain of FluPolA, vRNP-resident FluPolA carries out transcription primed by host capped RNAs[5,6]. Translation of viral proteins takes place in the cytoplasm, and newly synthesised FluPolA and NP proteins are imported into the nucleus to support replication. We propose that ANP32 proteins act as chaperones for free FluPolA as it enters the cell nucleus, by binding on the PA-C domain. The FluPolA–ANP32 complex cannot dock onto the RNAP II CTD, preventing competition with vRNP-resident FluPolA. Through interactions of ANP32B residues E151, E154 and P156 with FluPol$_E$ PB2$^{627}$ domain residues K627 and R630, FluPolA adopts an

encapsidating conformation (FluPol$_E$). A replication platform is assembled when the FluPol$_E$–ANP32 complex interacts with a vRNP-bound FluPolA, stabilising it in a replication-competent conformation (FluPol$_R$). As demonstrated by Zhu et al.[19] and others[10,11], ANP32B (or ANP32A) is an essential component of the influenza virus replication machinery, alongside NP and free FluPolA. In the context of the replication platform, FluPol$_E$ does not act catalytically but initiates cRNP assembly by capturing the 5' end of the nascent replication product, while ANP32 promotes the recruitment of NP to the nascent RNA, as proposed previously[15]. Our data suggest that mammalian adaptations not only allow the recruitment of mammalian ANP32 proteins, but also calibrate the different FluPolA oligomers to ensure assembly of the replication platform, enhancing our understanding of the molecular mechanisms behind mammalian adaptations in avian-origin FluPolA.

## Methods

### Cells and plasmids

HEK 293T cells were maintained in Dulbecco's Modified Eagle Medium (Sigma) with 10% FCS (Sigma); human eHAP (Horizon Discovery) TKO cells were maintained in Iscove's Modified Eagle Medium (Sigma) with 10% FCS, at 37 °C in a 5% $CO_2$ atmosphere. ANP32A, ANP32B and ANP32E knockout was achieved using CRISPR/Cas9 technology; this is described in detail elsewhere[10,26]. Sf9 cells were maintained in Sf-900 II serum free medium (Gibco) at 27 °C without $CO_2$.

pCAGGS plasmids expressing H5N1 Tky05 vRNP components PB1, PB2, PA and NP are described elsewhere[55]. Mutagenesis to obtain PB1 K557E and PA Q556R was performed by overlapping PCR as described[26]. PB2 K627E and R630A constructs were made by mutagenesis PCR using primers GCTGCTCCTCCTGAGCAGTCCCGTATGC and GCATACGGGACTGCTCAGGAGGAGCAGC, and CCACCGAAGCAG AGCGCAATGCAGTTTTCTTCTC and GAGAAGAAAACTGCATTGCGCT CTGCTTCGGTG, respectively. ANP32B constructs pcDNA-ANP32B E151A and P156A were obtained by mutagenesis PCR using primers GGATACGATCGCGCGGACCAAGAGGC and GGCCTCTTGGTCCGCGC GATCGTATC, and CGAGGACCAAGAGGCCGCTGATTCCGATGCCG and CCTCGGCATCGGAATCAGCGGCCTCTTGGTCCTC, respectively.

### Protein expression and purification

FluPolA subunits PA 556R, PB1 577E and protein A-tagged PB2 of A/turkey/Turkey/1/2005 (H5N1), with or without human ANP32B, were expressed in Sf9 cells from codon-optimised genes (Synbio) cloned into a single baculovirus using the MultiBac system[56]. Mutations PB1 K577E and PA Q556R were introduced by site-directed PCR mutagenesis. Sf9 cell suspension cultures, maintained in Sf-900 II serum-free medium (Gibco) without antibiotics, were infected with baculovirus and incubated for 72 h at 27 °C with shaking. Cells were harvested by centrifugation and lysed by sonication (3 × 30 s with 30 s intervals) in buffer A (50 mM Hepes:NaOH (pH 7.5), 150 mM KCl, 10% ($v/v$) glycerol, 0.05% ($w/v$) octylthioglucoside, 1 mM DTT), complemented with protease inhibitors (Roche, cOmplete Mini, EDTA-free) and 100 µg/ml RNase A. The lysate was clarified by centrifugation (35,000 × g, 45 mins, 4 °C) and the supernatant was incubated with IgG sepharose 6 Fast Flow (Cytiva) for 3 h at 4 °C. After binding, the beads were washed extensively with buffer A and FluPolA (+/− ANP32B) was released overnight at 4 °C with 0.5 mg tobacco etch virus protease in buffer A. The supernatant containing FluPolA (+/− ANP32B) was collected by centrifugation, and SEC was performed on a Superdex 200 Increase 10/300 GL column (GE Healthcare) in buffer B (25 mM Hepes:NaOH (pH 7.5), 150 mM KCl, 5% ($v/v$) glycerol, 1 mM DTT).

GST-tagged human ANP32B for pull-down assays was expressed in *E. coli* overnight at 18 °C in LB growth medium, upon 1 mM IPTG induction at $OD_{600} = 0.6$. Cells were lysed by sonication (3 × 45 s with 30 s intervals) in buffer C (50 mM Hepes:NaOH (pH 7.5), 150 mM NaCl, 10% ($v/v$) glycerol, 0.05% ($w/v$) octylthioglucoside, 1 mM DTT), complemented with protease inhibitors (Roche, cOmplete Mini, EDTA-free), 100 µg/ml RNase A and lysozyme (Sigma). The lysate was clarified by centrifugation (35,000 × g, 45 min, 4 °C) and the supernatant was incubated with glutathione sepharose™ 4B (Cytiva) for 3 h at 4 °C with rotation. After binding, the beads were washed extensively with buffer C and the proteins were released overnight at 4 °C with PreScission human rhinovirus 3 C protease in buffer C. The supernatant containing the protein was collected and SEC was performed on a Superdex 200 Increase 10/300 GL column (GE Healthcare) in buffer D (25 mM Hepes:NaOH (pH 7.5), 150 mM NaCl, 5% ($v/v$) glycerol, 1 mM DTT).

### Cryo-EM sample preparation

Grids were prepared using a Vitrobot mark IV (Thermo Fisher Scientific) at 100% relative humidity. Quantifoil Holey Carbon R2/1 200 mesh gold grids were glow discharged, before applying 3.5 µl of sample at around 0.3 mg/ml and blotted for 5 s, blot force 15, before vitrification in liquid ethane.

### Cryo-EM image collection

Cryo-EM data were collected at the Oxford Particle Imaging Centre (OPIC), on a 300 kV G3i Titan Krios microscope (Thermo Fisher Scientific) fitted with a SelectrisX energy filter and Falcon IV direct electron detector. Automated data collection was set up in EPU 3.4 and movies were recorded in EER format. Data were collected using AFIS with a total dose of ~50 e⁻/Å², a calibrated pixel size of 0.932 Å/pix and a 10 eV slit. For the FluPolA–ANP32B dataset 14,937 movies were collected without tilt and 5397 movies were collected with the specimen tilted by 30°. For the FluPolA dimer–ANP32B dataset, 3985 movies were collected without tilt and 7305 movies were collected with the specimen tilted by 30° to improve the angular distribution of the particles. Sample-specific data collection parameters are summarised in Supplementary Table 1.

### Cryo-EM data processing

All datasets were processed using CryoSPARC V-4.2[57], following the workflow illustrated in Supplementary Fig. 2. The EER format movies were fractionated in 60 frames without applying an up-sampling factor. Pre-processing was performed using patch motion correction and patch-CTF estimation with default settings. Corrected micrographs with poor statistics where manually curated. A first round of blob picking followed by a round of 2D classification generated initial templates that were used for template picking. After 2D classification using 200 classes, 500 particles per batch size and 40 iterations, well-resolved classes were selected and three ab initio models were generated and further refined using heterogenous refinement.

For the monomeric FluPolA–ANP32B dataset, particles belonging to high-resolution maps were used to train a Topaz model and pick a new set of particles. These particles were 2D classified using 200 classes, 500 particles per batch size and 40 iterations and the selected particles were used to generate three new ab initio models that were then refined using heterogenous refinement. A single class was selected and the 1 M particles were refined using NU-refinement with per-particles CTF refinement. 3D classification without alignment, using ten classes and default settings led to two interesting classes after visual inspection, that were selected and independently refined using local NU-refinement. The first map containing 63k particles was refined to 3.12 Å resolution and had density for FluPolA core and ANP32B. The other class contained 110,000 particles and also led to a 3.12 Å resolution map after local NU-refinement. The FluPolA core, as well as $PA^{Endo}$ and the $PB2^{NLS}$ were resolved in the density but without density for ANP32B. Based on the orientation of the $PA^{Endo}$ and the $PB2^{NLS}$ domains, this conformation is analogous to the replicase.

For the dimeric FluPolA–ANP32B dataset, particles belonging to well-resolved classes were used to train a Topaz model (default settings) and pick a new set of particles. To recover rare views that could have been classified out during 2D classification, particles from Topaz picker were directly 3D classified using heterogenous refinement with all the models generated with ab initio duplicated. After three rounds of Topaz training/picking and heterogenous refinement, the class representing the replication platform, containing 245,000 particles, was refined using NU-refinement with per-particles CTF refinement. To address the intrinsic flexibility of the complex, a focused refinement was performed on the $FluPol_E + PB2^{627}$ domain of the $FluPol_R$ and ANP32B (Supplementary Fig. 2, blue dashed lines) and one on the $FluPol_R$ without the $PB2^{627}$ domain (Supplementary Fig. 2, green dashed lines). Each was independently 3D classified, without particle alignment and across ten classes using default settings. After visual inspection, in the case of $FluPol_R$ (without $PB2^{627}$), a lot of the particles lacked a well-ordered PB1 β-hairpin and PA arch domain[38], likely due to denaturation at the air-water interface. These particles were removed.

Regarding $FluPol_E + FluPol_R - PB2^{627} + ANP32B$, we did not observe any variation in the occupancy of ANP32B. The organisation of the PA-C and the $PB2^{627}$ domains from $FluPol_E$ and $FluPol_R$ are identical in the focus refinement and the consensus map. The 3D classification allowed us to select the two classes with the most ordered $PB2^{CBD}$, $PB2^{Mid-link}$ and $PA^{Endo}$ for model building. Each of them were selected and independently refined using local NU-refinement, leading to 3.2 Å maps for each of them. The focus-refined maps were combined using PHENIX combine focused map[58]. The combined map was then used for model building after assessing that the quality of the combined map was identical to the focused maps and that the interface was identical to the consensus map.

### Structure determination and model refinement

Initial modelling was performed using PDB 6RR7 by first fitting the entire FluPolA or individual domains. The ANP32B structure was predicted by AlphaFold2[59] implemented in CollabFold[60]. Fitting was done using UCSF ChimeraX[61,62]. In WinCoot 0.9.8.7[63], the restraints module was used to generate restraints at 4.3 Å and allow flexible refinement to fit the main chain into density. Multiple cycles of manual adjustment in WinCoot, followed by real space refinement in PHENIX[64], were used to improve model geometry. The final model geometry and map-to-model comparison were validated using PHENIX MolProbity[65]. All the map and model statistics are detailed in Supplementary Table 1. Structural analysis and figures were prepared using UCSF ChimeraX.

### RNAP II CTD binding competition assay

We employed a method previously developed in our laboratory and described in detail elsewhere[6]. In short, 20 μg of biotinylated RNAP II CTD peptide (four heptad repeats[6]) was incubated for 30 min at 4 °C with 10 μl of Pierce streptavidin magnetic beads in buffer E, containing 10 mM HEPES (PAA Laboratories), 150 mM NaCl, 0.1% NP40 and 1 mM PMSF. As described by Martínez-Alonso et al. [6], the CTD peptides had been chemically synthesised by solid-phase peptide synthesis and quality controlled by mass spectrometry by Cambridge Peptide Ltd. Following three washes in buffer E the beads were incubated with 1% BSA in buffer E for 45 min at 4 °C in order to reduce non-specific binding. Following a wash in buffer E, 3.5 ug of purified Tky05 FluPolA (PB1 577E and PA 556R), in the presence or absence of a threefold molar excess of ANP32B, was incubated with the beads for 1 h at 4 °C. Following five washes in buffer E, bound proteins were released by boiling in SDS sample buffer and separated on a 10% SDS-PAGE gel. Proteins were visualised by silver staining using the SilverXpress LC6100 kit (Invitrogen). Band intensity was quantitated using Image J and analysed in GraphPad Prism 10.

### vRNP reconstitution assays

HEK 293T or eHAP TKO cells were seeded in 24-well plates and transfected with pCAGGS plasmids encoding Tky05 PB1 577E, PA 556R, PB2 and NP (0.05 μg each), alongside the viral reporter plasmid pPolI-firefly luciferase[66] and ANP32B-encoding or empty pcDNA plasmids as indicated, using Lipofectamine™ 3000 (ThermoFisher) transfection reagent, with 2 μl P3000™ Enhancer Reagent and 3 μl Lipofectamine™ 3000 Reagent per μg plasmid DNA. Twenty-four hours after transfection cells were lysed in 60 μl Reporter Lysis Buffer for 30 min at room temperature with gentle shaking. Bioluminescence was then measured on a FLUOstar Omega plate reader (BMG Labtech), using the Luciferase Assay System (Promega).

### Immunoblotting

Exogenously expressed proteins ANP32B and PB2, and the cellular control protein Vinculin were probed with primary antibodies rabbit anti-ANP32B (Abcam, ab200836; 1/2000), rabbit anti-influenza A virus PB2 (Genetex, GTX125925; 1/2000) and rabbit anti-Vinculin (Abcam, ab129002; 1/2000), respectively. The backsides of the primary antibodies were then detected with goat anti-rabbit secondary antibody conjugated to horseradish peroxidase (Genetex, GTX213110-01; 1/10,000). Detection was carried out using either Cytiva ECL Start Western Blotting Detection reagent (Amersham) or Immobilon Western chemiluminescent HRP substrate (Millipore).

### Reporting summary

Further information on research design is available in the Nature Portfolio Reporting Summary linked to this article.

### Data availability

All data are included in the paper, supplementary information or source data; source data are provided with this paper. Structural data generated in this study have been deposited in PDB and EMDB under accession codes PDB 8R1L, EMD-18822 (monomeric FluPolA–ANP32B), PDB 8R1J EMD-18818 (dimeric FluPolA–ANP32B), EMD-18819 (consensus map), EMD-18820 ($FluPol_E + FluPol_R - PB2^{627}$ focused map) and EMD-18821 ($FluPol_R$ focused map). Structural data used in this study are available in the PDB database under accession codes 6FHH, 6XZR and 6RR7. Source data are provided with this paper.

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

## Acknowledgements

We thank Imre Berger for the MultiBac system. We thank Stephen Cusack and members of the Grimes, Fodor and Barclay Laboratories for helpful comments and discussions. We thank Jaime Evans for proofreading the main text. This work was supported by Wellcome Investigator Awards 200835/Z/16/Z and 222510/Z/21/Z (to J.M.G.) and Medical Research Council (MRC) programme grants MR/R009945/1 and MR/X008312/1 (to E.F.). O.C.S., C.M.S. and W.S.B. were supported by Wellcome grant P65286. Access to electron microscopes was provided by the OPIC Electron Microscopy Facility (funded by Wellcome JIF (060208/Z/00/Z) and equipment (093305/Z/10/Z) grants). Access to computational resources was supported by the Wellcome Trust Core Award Grant Number 203141/Z/16/Z with additional support from the NIHR Oxford BRC. Molecular graphics and analyses were performed with UCSF Chimera, developed by the Resource for Biocomputing, Visualization, and Informatics at the University of California, San Francisco, with support from NIH P41-GM103311. The views expressed are those of the author(s) and not necessarily those of the NHS, the NIHR or the Department of Health.

## Author contributions

E.S., L.C., W.S.B., J.M.G. and E.F. conceived and designed the study. E.S., L.C., J.R.K. and H.F. generated recombinant baculoviruses, purified protein and performed structural analyses. E.S., O.C.S. and C.M.S. generated plasmids and performed functional assays. E.S., L.C., E.F and J.M.G wrote the paper with input from all authors.

## Competing interests

The authors declare no competing interests.
