## [Peer Review File · Nature Communications]

Structures of H5N1 influenza polymerase with ANP32B reveal mechanisms of genome replication and host adaptationReviewers' Comments:

Reviewer #1:

Remarks to the Author:

The manuscript by Staller and Carrique et al. explores the interaction of acidic nuclear phosphoprotein 32 (ANP32) proteins, specifically ANP32B, on the RNA polymerase of avian influenza virus A (FluPolA), the enzyme that transcribes and replicates the viral RNA genome. They mutated the PB1 and PA domains of FluPolA to stabilize a complex of FluPolA and ANP32B that they further characterized structurally using single particle cryo-EM. Two major classes were identified in the cryo-EM analysis: a monomer of H5N1 FluPolA bound to human ANP32B and a dimer of H5N1 FluPolA complexed with ANP32B. The monomeric structure reveals how the binding of ANP32B to FluPolA would compete with binding to the C-terminal domain of the host RNAP II and allow for association with another FluPolA molecule to generate a replication platform. The dimeric structure reveals architecture of the replication platform of influenza A virus (IVA) and how it contrasts the conformation of the replication platform in avian influenza C (IVC) (Carrique, Fan, Walker, Keown, et al., Nature, 2020) and enables mapping of FluPolA adaptive mutations to the interface of the dimers.

The manuscript is well written, and the structural biology experiments are sound. This work builds upon the previous structural work on the H5N1 FluPolA (Fan, Walker, Carrique, Keown, et al., Nature, 2019) and IVC replication platform (Carrique, Fan, Walker, Keown, et al., Nature, 2020) shedding light on the IVA replication platform, which would be of interest to the influenza transcription/replication field. There are a several concerns that need to be addressed before consideration for publication in Nature Communications.

MAJOR COMMENTS:

-Page 3-4, Lines 68-81. The authors engineered two mutations to the Tky05 FluPolA for structural studies: PB1-K577E to disrupt the symmetric dimer interface and PA Q556R to enhance binding to ANP32B. While the cryo-EM suggest that the proteins are soluble and well-folded, is this mutant polymerase capable of binding template RNA and synthesizing cRNA and vRNA? What is the specific activity of this polymerase construct compared to wildtype? These are important controls to include since the authors are introducing mutations to stabilize a conformation of FluPolA with ANP32B that they claim to have functional significance.

-Page 5, Line 108-113. Authors show that ANP32B can outcompete RNAP II for binding to FluPolA. The experiment shown in Fig. 1e tests FluPolA binding with immobilized peptides, but the methodological details are not clear. Could the author clarify which Tky05 FluPolA construct they are using for this experiment (WT vs the PB1-K577E+PA=Q556R double mutant)? This is What are the peptides used for the pulldown experiments (e.g. unphos, scrambled)? How were these peptides prepared and how was the level of phosphorylation accessed/quantitated for S2P and S5P? In the no peptide control, there is signal in the gel for FluPolA binding yet the bar graph indicates no binding. How was the binding data quantitated and normalized? Additionally, key controls are missing in the gel such as ANP32B alone without FluPolA. FluPolA was incubated with threefold molar excess of ANP32B. How was this molar ration establish and how does it compared to cellular levels of FluPolA, RNAP II and ANP32B during infection?

-Page 5-6, Lines 115-144. Authors claim that the conical replication complex for IAV is composed of an asymmetric dimer of FluPolA and ANP32B. Since FluPolA readily forms symmetric and asymmetric oligomers, what is the role of ANP32B in the context of the FluPolA replication platform once it is formed? In the dimer structure, ANP32B interacts exclusively with the FluPolE PB2627 domain, with no observable interactions with FluPolR. This configuration contrasts the IVC replication platform in which ANP32A seems to bind both FluPolC protomers contributing to dimerization and activity of the

replication platform. How is asymmetric dimerization of FluPoIA established and is this tied to binding of ANP32B? How does ANP32B affect the transcription activity of the IVA replication complex?

-Extended Data Fig. 2. The composite map of the replication platform is derived from independent maps that have been generated from two separate 3D classifications. While this processing strategy gives an overall improved cryo-EM map, it can be misleading and problematic since the consensus particles are split into two independent halves, signal subtracted and are sorted as individual rigid bodies. Thus, heterogeneity is only monitored in one half of the complex while ignoring the other half. Could the authors elaborate on the conformational heterogeneity seen in the 10 classes from the 3D classification for both FluPoIE+FluPoIR-PB627 and FluPoIR local refinements? What were the criteria for selecting the classes for subsequent refinement and composite map generation? What is the overlap between the 80k particles in the FluPoIE+FluPoIR-PB627 versus 105k particles in the FluPoIR subselections? By extension, is the conformation of FluPoIE+FluPoIR-PB627 the same between a full refinement of the complex for the 80k particles versus the 105k particles that make up the FluPoIR local refinement map? Likewise, how does FluPoIR compare between these populations of particles?

MINOR COMMENTS

-Page 6, Lines 125-126. Provide references for the published transcriptase and replicase structures.

-Pages 12-14, Cryo-EM data processing. The authors use extensive processing procedures to produce the high-resolution maps presented in the manuscript, which may be of interest to readers who are performing similar cryo-EM analyses. Could the authors please provide more details in terms of the parameters used for Topaz picking (e.g. model architecture, training rate), CTF refinement (e.g. Local and Global CTF), and 3D Classification (e.g. online expectation maximization). This is important to outline in the methods since some of the cryoSPARC modules are in beta version, such as 3D Classification without alignment.

-Figures detailing amino acid interactions should show the supporting cryo-EM densities (e.g. Fig. 1b-c, Fig. 2c-g, Fig. 4b-c) that were used to interpret the interactions. This makes it more accessible for readers who may not have the maps and models readily at hand. The legend for these figures should detail how the density was generated (e.g. contour level, zoning distance).

-Fig. 1e. The samples represented in the bar graph are not clearly labeled nor aligned with the lanes of the corresponding gel.

-Fig. 2h. The authors outline the putative path for the ANP32BLCAR over a positively charged groove created by the dimerization interface between FluPoIE and FluPoIR. Does low-pass/local resolution filtering reveal any low-resolution cryo-EM density that could correspond to the LCAR domain of ANP32B? Could AF2 predictions of this region in the complex shed light on the potential trajectory of the LCAR domain and how it would bind NP? On a related note, does this region contribute to the dimerization of FluPol and contribute to RNA synthesis during IVA replication?

-Extended Data Table 1. Map resolution for the FluPoIA dimer – huANP32B should be corrected from "3,24" to "3.24".

-Extended Data Fig. 1c. The rotated structure shown in the right panel is clipped at the bottom.

-Extended Data Fig. 2. Does the locally refined map on the bottom left (FluPoIE+FluPoIR-PB627) include ANP32B? If so, the label should be updated to include this.

Reviewer #2:

Remarks to the Author:

The study explores cryo-EM structures of monomeric and dimeric H5N1 FluPoIA in complex with human ANP32B, shedding light on potential mechanisms for mammalian adaptations in avian-origin FluPoIA.

Overall Comments:

While the manuscript provides some insights, addressing some key points is crucial for strengthening its validity and supporting conclusions. This reviewer recommends revisions and some additional experiments to improve the overall quality of this manuscript.

In their publication in Nature in 2020, the authors emphasized the crucial role of the ANP32 protein in bridging two polymerases to form a replication platform. The interaction of ANP32a with both polymerases was highlighted in that study. However, in the current investigation on influenza A virus, it is observed that the ANP32 protein interacts only with one polymerase. Moreover, this interaction has been artificially strengthened through mutations. This raises concerns regarding the proposed significance of ANP32 in mediating polymerase interactions, suggesting that the polymerase itself may inherently exhibit a strong tendency to form an asymmetric dimer, independent of ANP32 mediation. This is a big concern and constitutes the principal point of apprehension regarding the authors' theory.

The findings presented in the current study offer some insights into the molecular mechanisms underlying mammalian adaptations in avian-origin FluPoIA. However, when considering the work as a whole, it does not appear to represent a significant breakthrough in the field. Several aspects need verification and modification, and there are numerous areas that require attention before the conclusions can be considered fully supported.

1. Concerns about ANP32-Mediated Polymerase Interactions:

The discrepancy between the observed interactions of ANP32 with both polymerases in the previous Nature publication and the current study's findings, where ANP32 interacts with only one polymerase, needs to be thoroughly addressed. The impact of artificial mutations on this interaction and its implications for the overall model should be clarified.

2. The manuscript describes different conformations of FluPoIE, involving various interactions, particularly with PAN. The mention of PAN 51-72 loop stabilizing this conformation raises questions about the functional significance of this new conformation. The structural analysis indicates that PAN in FluPoIE does not participate in replication. It is suggested that this conformation might be a state in the protein folding process rather than having practical functional significance. Actually, ANP32 family proteins have been shown to function as chaperones (such as PMID: 24613878, PMID: 18039846). It is possible that this conformation is just a folding intermediate.

3. There is an absence of information in the entire manuscript regarding the density of key amino acid interactions and the quality assessment of the overall structure. This lack of data makes it challenging to ascertain the authenticity of the described interactions. It is crucial to provide evidence supporting the observed molecular interactions and to present an assessment of the overall structural quality to enhance the manuscript's credibility.

4. The only functional pulldown assay described in the manuscript is unclear, and there is a lack of proper labeling for statistical results. The absence of clear indications makes it difficult for readers to identify specific elements, raising concerns about the thoroughness of the manuscript's preparation.

5. The mutations purportedly affecting the influenza virus's host range lack a clear summary. The manuscript does not specify which positions and amino acid mutations influence what aspects, and whether these mutations exhibit any structural patterns. Without this information, predicting the future trends in virus spread based on the current structure becomes untenable.

6. Page 2, Line 27. The authors propose that "ANP32B serves as a chaperone, directing newly synthesized FluPoIA towards an RNA-associated FluPoIA molecule to form an asymmetric dimer."

However, the dimeric complex of FluPoIA with huANP32B lacks vRNA or cRNA. Given FluPol flexible conformation, depending on template RNA or host proteins, further functional assays, such as in vitro transcription and replication assays, are recommended to elucidate the role of ANP32B in replication platform formation.

7. Fig. 1e, Page 5: The ability of the FluPoIA to bind serine 5 phosphorylated (S5P) version of the CTD was significantly reduced in the presence of the host factor ANP32B. Does the binding competition assay using FluPoIA contain mutations PB1 K577E and PA Q556R? Since these mutations affect the oligomerization of the polymerase, one would expect that these mutations may have potential impact on the binding of ANP32A to polymerase without these mutations. It would be better to test the polymerase without these mutations as well, if any, and to measure their affinity values.

Some specific Points:

1. Fig. 1b: The side chain of K635 in Fig. 1b requires improvement for a clearer representation.

2. On Page 5, Line 121, and Page 6, Line 130, the "product exit channel" and "template exit channel" are mentioned. Labeling these channels in the corresponding figures (Fig. 1b and Fig. 1d) would enhance reader comprehension.

3. In Figure 2c, it is recommended to indicate the interface from FluPoIE or FluPoIR for better contextualization.

4. Page 6, Line 127. The expression "while in the present conformation..." is ambiguous. Providing PDB numbers and clarifying the statement would enhance clarity.

5. Please fix the bond angle outliers in the PDB validation report.

RESPONSE TO REVIEWERS' COMMENTS

Reviewer #1

The manuscript by Staller and Carrique et al. explores the interaction of acidic nuclear phosphoprotein 32 (ANP32) proteins, specifically ANP32B, on the RNA polymerase of avian influenza virus A (FluPolA), the enzyme that transcribes and replicates the viral RNA genome. They mutated the PB1 and PA domains of FluPolA to stabilize a complex of FluPolA and ANP32B that they further characterized structurally using single particle cryo-EM. Two major classes were identified in the cryo-EM analysis: a monomer of H5N1 FluPolA bound to human ANP32B and a dimer of H5N1 FluPolA complexed with ANP32B. The monomeric structure reveals how the binding of ANP32B to FluPolA would compete with binding to the C-terminal domain of the host RNAP II and allow for association with another FluPolA molecule to generate a replication platform. The dimeric structure reveals architecture of the replication platform of influenza A virus (IVA) and how it contrasts the conformation of the replication platform in avian influenza C (IVC) (Carrique, Fan, Walker, Keown, et al., Nature, 2020) and enables mapping of FluPolA adaptive mutations to the interface of the dimers.

The manuscript is well written, and the structural biology experiments are sound. This work builds upon the previous structural work on the H5N1 FluPolA (Fan, Walker, Carrique, Keown, et al., Nature, 2019) and IVC replication platform (Carrique, Fan, Walker, Keown, et al., Nature, 2020) shedding light on the IVA replication platform, which would be of interest to the influenza transcription/replication field. There are several concerns that need to be addressed before consideration for publication in Nature Communications.

We thank the Reviewer for their positive comments on our manuscript.

MAJOR COMMENTS:

-Page 3-4, Lines 68-81. The authors engineered two mutations to the Tky05 FluPolA for structural studies: PB1-K577E to disrupt the symmetric dimer interface and PA Q556R to enhance binding to ANP32B. While the cryo-EM suggest that the proteins are soluble and well-folded, is this mutant polymerase capable of binding template RNA and synthesizing cRNA and vRNA? What is the specific activity of this polymerase construct compared to wildtype? These are important controls to include since the authors are introducing mutations to stabilize a conformation of FluPolA with ANP32B that they claim to have functional significance.

We apologise for not making it clear that the structural studies were carried out with a FluPolA derived from a viable virus that is capable of transcription and replication, as well as multiple rounds of infection in human cells and mice. The behaviour of this virus and its polymerase are described at length in Sheppard et al Nat Commun 2023; these authors use a wide variety of *in vivo* and *in vitro* assays to characterize the activity of this polymerase.

Briefly, the FluPolA we use is derived from the avian H5N1 strain A/turkey/Turkey/1/2005 (Tky05) which was passaged in human cells lacking ANP32A and ANP32B (dKO) and acquired two mutations – PB1 K577E and PA Q556R – which allowed it co-opt an alternative host factor, ANP32E, to support its replication. Importantly, as shown in Sheppard et al Nat

Commun 2023, this mutant polymerase can still be supported by ANP32B, underpinning the functional relevance of our structure.

We would also like to emphasize that the PB1 K577E / PA Q556R pair of adaptations evolved previously in an avian-origin H3N2 virus adapting to mice (Ping et al PLoS One 2011). More generally, it has been established that mutations of the lysines at positions 577 and 578 of the PB1 subunit arise when avian-origin influenza A viruses adapt to human cells or mice, i.e. suboptimal ANP32 proteins lacking the 33-amino acid avian ANP32A insertion. These substitutions consistently lead to a reduction in symmetric dimer formation, accompanied by an increase in FluPol activity (Sheppard et al Nat Commun 2023; Chen et al PLoS Path 2019; Kamiki et al Viruses 2018; Gunl et al Nat Commun 2023).

The Q556R substitution in the PA subunit has also been described, including recently in an H9N2 virus adapting to *Anp32A* gene-edited chickens lacking the proviral ANP32A signature 129N/130D (Idoko-Akoh Nat Commun 2023). PA Q556R also commonly evolves in avian-origin influenza A viruses adapting to mice (Brown et al PNAS 2001; Choi et al Sci Rep 2017; Xiang et al J Infect Dis 2018; Wasik et al J Virol 2019).

We have now included a new section in the Introduction providing a detailed introduction to the polymerase we used in the structural analysis, and the significance of the PB1 K577E / PA Q556R adaptive changes. We extensively refer to the previous publication Sheppard et al Nat Commun 2023 that evaluates the impact of these adaptive changes on both polymerase activity and the phenotype of the virus.

-Page 5, Line 108-113. Authors show that ANP32B can outcompete RNAP II for binding to FluPolA. The experiment shown in Fig. 1e tests FluPolA binding with immobilized peptides, but the methodological details are not clear. Could the author clarify which Tky05 FluPolA construct they are using for this experiment (WT vs the PB1-K577E+PA=Q556R double mutant)? This is What are the peptides used for the pulldown experiments (e.g. unphos, scrambled)? How were these peptides prepared and how was the level of phosphorylation accessed/quantitated for S2P and S5P? In the no peptide control, there is signal in the gel for FluPolA binding yet the bar graph indicates no binding. How was the binding data quantitated and normalized? Additionally, key controls are missing in the gel such as ANP32B alone without FluPolA. FluPolA was incubated with threefold molar excess of ANP32B. How was this molar ration establish and how does it compared to cellular levels of FluPolA, RNAP II and ANP32B during infection?

We apologise for the lack of methodological detail. Here we use an assay previously established in our lab to evaluate the binding of FluPolA to the RNAP II CTD (Martínez-Alonso et al J Virol 2016; Keown et al Nat Commun 2022). In Fig. 1e, we use FluPolA with PB1 K577E and PAQ556R as in the structural analysis. The peptide design was described in Martínez-Alonso et al J Virol 2016; they were chemically synthesised by solid-phase peptide synthesis and quality controlled by mass spectrometry by Cambridge Peptide Ltd. (Martínez-Alonso et al J Virol 2016). We now provide a more detailed description of the peptides in the Methods, including references to the previously published studies.

Band intensity was quantitated using Image J and analysed in GraphPad Prism 10. The signal in the no peptide lane was set to zero and subtracted from the other lanes, essentially rendering background signal (i.e. FluPolA binding to sepharose beads in the absence of CTD peptide) zero. We now describe the quantitation in detail in the Methods and figure legends.

We believe a control with ANP32B on its own, without FluPol, would not provide additional information. If ANP32B were to interact with the CTD peptide by itself, thus blocking access for FluPol, we would see ANP32B on the gels in the S5P + ANP32B lane (Fig. 1e), especially given that it is present in excess compared to FluPol. Prior to performing the experiment shown in Fig. 1e, we performed a titration experiment using increasing amounts of ANP32B to establish suitable relative amounts of the proteins used in the assay. We now include this titration in the Supplementary Information as Supplementary Fig. 3. In this experiment we add up to 10-fold molar excess of ANP32B over FluPol but no ANP32B is observed in the pulldown, confirming ANP32B does not bind the CTD peptides (Supplementary Fig. 3). Please note that it is difficult if not impossible to mimic FluPolA, RNAP II and ANP32B levels in a cell during infection as relative protein levels change over time, with the levels of FluPolA increasing while the large subunit of RNAP II is being degraded (Vreede et al Virology 2010; Rodriguez et al J Virol 2007).

-Page 5-6, Lines 115-144. Authors claim that the conical replication complex for IAV is composed of an asymmetric dimer of FluPolA and ANP32B. Since FluPolA readily forms symmetric and asymmetric oligomers, what is the role of ANP32B in the context of the FluPolA replication platform once it is formed? In the dimer structure, ANP32B interacts exclusively with the FluPolE PB2627 domain, with no observable interactions with FluPolR. This configuration contrasts the IVC replication platform in which ANP32A seems to bind both FluPolC protomers contributing to dimerization and activity of the replication platform. How is asymmetric dimerization of FluPolA established and is this tied to binding of ANP32B? How does ANP32B affect the transcription activity of the IVA replication complex?

In our hands, purified FluPolA readily forms symmetric but not asymmetric dimers. Our model suggests that binding of ANP32B to newly synthesised (i.e. RNA-free) FluPol leads to a conformational shift to an encapsidating FluPol (FluPolE), specifically through interactions of ANP32B residues E151, E154 and P156 with FluPolE PB2⁶²⁷ domain residues K627 and R630. The biological relevance of these interactions is borne out by vRNP reconstitution assays that we have included in the revised version of our manuscript (Supplementary Fig. 4). In the context of the replication platform, ANP32B promotes the recruitment of NP to the nascent RNA, as we proposed previously (Wang et al NAR 2022). As demonstrated in Zhu et al (Plos Biol 2023) and elsewhere, ANP32B (or ANP32A) is an essential component of the influenza virus replication machinery, alongside NP and free FluPol.

We have extended our Discussion to clarify this by presenting our model in the context of previously published studies. We have also included new experimental data demonstrating the functional importance of FluPolE-ANP32B interactions (Supplementary Fig. 4).

-Extended Data Fig. 2. The composite map of the replication platform is derived from independent maps that have been generated from two separate 3D classifications. While this processing strategy gives an overall improved cryo-EM map, it can be misleading and problematic since the consensus particles are split into two independent halves, signal subtracted and are sorted as individual rigid bodies. Thus, heterogeneity is only monitored in one half of the complex while ignoring the other half. Could the authors elaborate on the conformational heterogeneity seen in the 10 classes from the 3D classification for both FluPolE+FluPolR-PB627 and FluPolR local refinements? What were the criteria for selecting the classes for subsequent refinement and composite map generation? What is the overlap

between the 80k particles in the FluPol_E+FluPol_R-PB627 versus 105k particles in the FluPol_R subselections? By extension, is the conformation of FluPol_E+FluPol_R-PB627 the same between a full refinement of the complex for the 80k particles versus the 105k particles that make up the FluPol_R local refinement map? Likewise, how does FluPol_R compare between these populations of particles?

The cryo-EM data processing strategy used in this study is an established one, especially for big complexes such as polymerases (Vergara-Cruces et al Cell 2024). Combining focused refinement and 3D classifications allowed us to tackle inherent flexibility of the different components of the complex. Based on the first consensus map, we could identify that the replication platform had two major flexible halves: 1) FluPol_E + FluPol_R-PB2⁶²⁷ + ANP32B, and 2) FluPol_R (without PB2⁶²⁷). A local refinement of each of these halves gave the biggest improvement in terms of map quality for interpretation and model building. However, 3D classification allowed us to further remove heterogeneity. In the case of FluPol_R (without PB2⁶²⁷), a lot of the particles lacked a well-ordered PB1 β-hairpin and PA arch domain, likely due to denaturation at the air-water interface. These particles were removed during 3D classification. Regarding the FluPol_E+FluPol_R-PB2⁶²⁷+ANP32B, we did not observe any variation in the occupancy of ANP32B. The organization of the PA-C and the PB2⁶²⁷ domains from FluPol_E and FluPol_R are identical in the focus refinement and the consensus map. The 3D classification allowed us to select the classes with the most ordered PB2^{CBD}, PB2^{Mid-link} and PA^{Endo} for model building.

The number of particles overlapping between the two final classes that were used to generate the final composite map is 35,023. Doing a final refinement of the entire replication platform with this particle set, or the set of particles selected after each 3D classifications, led to an overall replication platform similar to the consensus map. However, the flexible parts initially enriched through 3D classification become blurry again in the other half that has not been classified (or in both halves in the case of the overlap set). In addition, the resolution and interpretability of the map compared to the consensus refinement decrease significantly as expected.

To clarify these points in the cryo-EM processing strategy we have now included further detail in the Methods.

MINOR COMMENTS

-Page 6, Lines 125-126. Provide references for the published transcriptase and replicase structures.

We now provide references (Kouba et al NSMB 2019, Fan et al Nature 2019, Carrique et al Nature 2020).

-Pages 12-14, Cryo-EM data processing. The authors use extensive processing procedures to produce the high-resolution maps presented in the manuscript, which may be of interest to readers who are performing similar cryo-EM analyses. Could the authors please provide more details in terms of the parameters used for Topaz picking (e.g. model architecture, training rate), CTF refinement (e.g. Local and Global CTF), and 3D Classification (e.g. online expectation maximization). This is important to outline in the methods since some of the cryoSPARC modules are in beta version, such as 3D Classification without alignment.

We have modified the methods section to specify that default settings were used.

-Figures detailing amino acid interactions should show the supporting cryo-EM densities (e.g. Fig. 1b-c, Fig. 2c-g, Fig. 4b-c) that were used to interpret the interactions. This makes it more accessible for readers who may not have the maps and models readily at hand. The legend for these figures should detail how the density was generated (e.g. contour level, zoning distance).

We would prefer to leave the figures unchanged as the addition of the cryo-EM density to figure panels displaying large interaction areas will make the figures heavy and will not serve the reader (see example below). In addition, interpretation of the contacts is mainly done through automatic search of the contact distance within the model after model building and is independent of the map, which is just used as an assessment of the quality of the model building. Moreover, the density map contour levels are not relevant for cryo-EM as the map is on an arbitrary scale and, as global sharpening is applied, all the areas of a map are not made to be displayed at the same threshold.

-Fig. 1e. The samples represented in the bar graph are not clearly labelled nor aligned with the lanes of the corresponding gel.

We thank the Reviewer for pointing out this error; we have now fixed this.

-Fig. 2h. The authors outline the putative path for the ANP32BLCAR over a positively charged groove created by the dimerization interface between FluPolE and FluPolR. Does low-pass/local resolution filtering reveal any low-resolution cryo-EM density that could correspond to the LCAR domain of ANP32B? Could AF2 predictions of this region in the complex shed light on the potential trajectory of the LCAR domain and how it would bind NP? On a related note, does this region contribute to the dimerization of FluPol and contribute to RNA synthesis during IVA replication?

Careful analysis of the EM maps in addition to extensive 3D classifications focusing on the positively charged groove did not reveal extra density for the LCAR. Unfortunately, multiple attempts at using AF2 provided no further insights either. The LCAR domain is intrinsically disordered. AF2 also failed to correctly predict the interaction between FluPol_E PB2^{627-NLS} and FluPol_R PB2⁶²⁷ with or without ANP32B, due to the limited contact area between them. Prediction of the FluPol complex is computationally demanding and challenging due to the multiple conformations that can be adopted by FluPol and was also unsuccessful in our hands.

Regarding the binding to NP, as mentioned above, our previous study (Wang et al NAR 2022) demonstrates that NP is recruited to the replication complex through the ANP32 LCAR domain that binds the RNA binding groove of NP. Based on the position of the ANP32B LRR domain in our structure and the visible part of the LCAR, it is likely that the LCAR extends into the positively charged groove formed by the FluPol_E PB2^{627-NLS} and FluPol_R PB2⁶²⁷ domains, and therefore contributes to the stability of the replication platform and, by extension, RNA synthesis. In support of this, we show that truncations of the LCAR prevent viral RNA replication (Zhu et al PLoS Biol 2023).

We have extended our Discussion to address these issues in more detail.

-Extended Data Table 1. Map resolution for the FluPolA dimer – huANP32B should be corrected from “3,24” to “3.24”.

We have corrected this typo.

-Extended Data Fig. 1c. The rotated structure shown in the right panel is clipped at the bottom.

Thank you for highlighting this issue; this figure panel has been removed from the revised version of the manuscript.

-Extended Data Fig. 2. Does the locally refined map on the bottom left (FluPol_E+FluPol_R-PB627) include ANP32B? If so, the label should be updated to include this.

We thank the Reviewer for pointing this out; we have updated the label as ANP32B was indeed included in the locally refined map and 3D classification.

Reviewer #2

The study explores cryo-EM structures of monomeric and dimeric H5N1 FluPolA in complex with human ANP32B, shedding light on potential mechanisms for mammalian adaptations in avian-origin FluPolA.

Overall Comments:

While the manuscript provides some insights, addressing some key points is crucial for strengthening its validity and supporting conclusions. This reviewer recommends revisions and some additional experiments to improve the overall quality of this manuscript. In their publication in Nature in 2020, the authors emphasized the crucial role of the ANP32 protein in bridging two polymerases to form a replication platform. The interaction of ANP32a with both polymerases was highlighted in that study. However, in the current investigation on influenza A virus, it is observed that the ANP32 protein interacts only with one polymerase. Moreover, this interaction has been artificially strengthened through mutations. This raises concerns regarding the proposed significance of ANP32 in mediating polymerase interactions, suggesting that the polymerase itself may inherently exhibit a strong tendency to form an asymmetric dimer, independent of ANP32 mediation. This is a big concern and constitutes the principal point of apprehension regarding the authors' theory.

The findings presented in the current study offer some insights into the molecular mechanisms underlying mammalian adaptations in avian-origin FluPolA. However, when

considering the work as a whole, it does not appear to represent a significant breakthrough in the field. Several aspects need verification and modification, and there are numerous areas that require attention before the conclusions can be considered fully supported.

We thank the Reviewer for their comments. As we explain below in response to point 1, there is indeed a difference in how ANP32A interacts with FluPolC in our previous study and how ANP32B interacts with FluPolA in our current study. We would like to emphasize that influenza A and C viruses have diverged significantly over time, encoding a polymerase that is only about 20-40 % identical at the amino acid level (see Extended Data Table 2 in Hengrung et al Nature 2015). Therefore, it is not surprising that they exhibit some differences in ANP32 protein binding. Please note that FluPolA and FluPolC also exhibit differences in Pol II CTD binding, equally involving the PA-C domain (see Krischuns et al PLoS Path 2022 and Walker and Fodor Trends Microbiol 2019 for a review).

We apologise for the potentially misleading wording that led the Reviewer to believe that we artificially strengthened the interaction with ANP32B. As we explain above, in response to the first point of Reviewer 1, the FluPolA we use for the structural studies is derived from a viable virus, described comprehensively in Sheppard et al Nat Commun 2023. This FluPolA, when purified in the absence of ANP32B, preferentially forms monomeric heterotrimers (see Fig. 5D in Sheppard et al Nat Commun 2023).

We have now inserted a section in the Introduction where we explain the origin of FluPolA that we use and also describe its characteristics in more detail, referring extensively to Sheppard et al Nat Commun 2023.

1. Concerns about ANP32-Mediated Polymerase Interactions:

The discrepancy between the observed interactions of ANP32 with both polymerases in the previous Nature publication and the current study's findings, where ANP32 interacts with only one polymerase, needs to be thoroughly addressed. The impact of artificial mutations on this interaction and its implications for the overall model should be clarified.

As stated above, the polymerases used in our previous and current studies are derived from influenza C and A virus, respectively. These polymerases share only 20-40 % sequence identity (see Extended Data Table 2 in Hengrung et al Nature 2015). Hence, it is not entirely surprising they show differences in ANP32 binding. In fact, the discovery of these differences is one of the strengths of our study, which is why we report it under the heading “Comparison of the IAV and ICV replication platforms”. As stated, PB1 K577E and PA Q556R are not artificial mutations; they derive from an infectious virus. The effect of these mutations, which are known to occur naturally upon avian to mammalian adaptation (Ping et al PLoS One 2011; Chen et al PLoS Path 2019; Kamiki et al Viruses 2018; Idoko-Akoh et al Nat Commun 2023; Brown et al PNAS 2001; Choi et al Sci Rep 2017; Xiang et al J Infect Dis 2018; Wasik et al J Virol 2019), including their effect on polymerase activity, ANP32 binding, and polymerase dimerisation, have been thoroughly evaluated in Sheppard et al Nat Commun 2023.

We have now revised the text, explaining the origin of the FluPolA we use, and also describe its characteristics in more detail, referring to Sheppard et al Nat Commun 2023.

2. The manuscript describes different conformations of FluPolE, involving various interactions, particularly with PAN. The mention of PAN 51-72 loop stabilizing this

conformation raises questions about the functional significance of this new conformation. The structural analysis indicates that PAN in FluPol_E does not participate in replication. It is suggested that this conformation might be a state in the protein folding process rather than having practical functional significance. Actually, ANP32 family proteins have been showed to function as chaperones (such as PMID: 24613878, PMID: 18039846). It is possible that this conformation is just a folding intermediate.

We apologise for the unclear wording of the section describing FluPol_E. We do not describe different conformations of FluPol_E, rather we compare the FluPol_E conformation, which is novel, to published transcriptase (FluPol_T) and replicase (FluPol_R) conformations (Supplementary Fig. 1b). FluPol_T is responsible for generating capped RNA-primed viral mRNA, while FluPol_R replicates the viral genome in a primer independent manner. Although FluPol_E is catalytically inactive, it does participate in genome replication by playing a role in the formation of the replication platform. It likely contributes to the stabilization of the FluPol_R conformation and captures the 5' end of the nascent replication product, initiating its assembly into a ribonucleoprotein complex with viral NP (see Wang et al NAR 2022). As such, rather than being a folding intermediate, FluPol_E is indispensable for viral genome replication. Previously published data implicating the PA⁵¹⁻⁷² loop in viral genome replication but not transcription are fully consistent with our observation here that this PA loop contributes to the stabilization of the FluPol_E conformation. We are aware that ANP32 proteins, among many other things, function as histone chaperones; indeed we propose a chaperone function.

We have revised the section describing the FluPol_E conformation and state that without PA loop 51-72 the FluPol_E configuration cannot form, and without FluPol_E the replication platform cannot form. We also modified the Discussion to describe the role of FluPol_E in viral genome replication in more detail.

3. There is an absence of information in the entire manuscript regarding the density of key amino acid interactions and the quality assessment of the overall structure. This lack of data makes it challenging to ascertain the authenticity of the described interactions. It is crucial to provide evidence supporting the observed molecular interactions and to present an assessment of the overall structural quality to enhance the manuscript's credibility.

As we have mentioned in response to a comment from Reviewer 1, we think that the addition of the cryo-EM density to figure panels displaying large interaction areas would make the figures heavy and would not serve the reader. We believe that the readers who are interested in the quality of the map will benefit from opening the cryo-EM maps and models in freely available software such as Coot and ChimeraX or even from the PDB itself to visualize the map in 3D as opposed to 2D snapshots.

4. The only functional pulldown assay described in the manuscript is unclear, and there is a lack of proper labeling for statistical results. The absence of clear indications makes it difficult for readers to identify specific elements, raising concerns about the thoroughness of the manuscript's preparation.

We apologise for the lack of information. We have added further information regarding the quantitation of the data in the figure legends and Methods and revised Fig. 1e (see also our response to a similar comment from Reviewer 1).

5. The mutations purportedly affecting the influenza virus's host range lack a clear summary. The manuscript does not specify which positions and amino acid mutations influence what aspects, and whether these mutations exhibit any structural patterns. Without this information, predicting the future trends in virus spread based on the current structure becomes untenable.

We provide a summary of mutations affecting the host range of influenza virus in Fig. 4a and Supplementary Table 2. We show in Fig. 4a where known FluPolA adaptations are positioned on our structure of the replication platform and observe that many of them cluster along the FluPol_R-FluPol_E dimer interface and/or the interface between FluPol_E and ANP32B. Their location is also specified in Supplementary Table 2, indicating whether they would be relevant in FluPol_R or FluPol_E, and at the FluPol_R-FluPol_E or FluPol_E-ANP32 interface, and we provide references to primary literature describing each mutation in detail. FluPol mutations affecting host range have been reviewed comprehensively elsewhere and we feel that discussing each mutation in detail is beyond the scope of this work (see for instance Gilbertson et al Current Opinion in Virology 2023).

We state in the revised version of our manuscript that mutations in the viral polymerase are vital to the process of host adaptation, particularly in the process of avian influenza A viruses adapting to mammalian hosts, and refer to the excellent review by Gilbertson et al 2023. We have revised the Discussion and no longer discuss the prediction of future trends.

6. Page 2, Line 27. The authors propose that "ANP32B serves as a chaperone, directing newly synthesized FluPolA towards an RNA-associated FluPolA molecule to form an asymmetric dimer." However, the dimeric complex of FluPolA with huANP32B lacks vRNA or cRNA. Given FluPol flexible conformation, depending on template RNA or host proteins, further functional assays, such as *in vitro* transcription and replication assays, are recommended to elucidate the role of ANP32B in replication platform formation.

Assays elucidating the role of ANP32B in replication platform formation using *in vitro* transcription and replication assays have recently been published from our lab (Zhu et al PLoS Biol 2023). In that study we use vRNPs derived from purified virions and show that the vRNP-associated FluPol is able to perform transcription (viral mRNA synthesis) in the presence of a source of capped RNA primer. However, for vRNPs to replicate, the addition of ANP32B along with free FluPolA and NP is essential. We also perform experiments demonstrating that both the N-terminal LRR domain and the N-terminal part of the C-terminal LCAR are essential for ANP32's ability to support influenza virus genome replication.

We now discuss the results of this paper in more detail in the revised version of our Discussion, further emphasising the importance of ANP32B for influenza virus genome replication.

7. Fig. 1e, Page 5: The ability of the FluPolA to bind serine 5 phosphorylated (S5P) version of the CTD was significantly reduced in the presence of the host factor ANP32B. Does the binding competition assay using FluPolA contain mutations PB1 K577E and PA Q556R? Since these mutations affect the oligomerization of the polymerase, one would expect that these mutations may have potential impact on the binding of ANP32A to polymerase without these mutations. It would be better to test the polymerase without these mutations as well, if any, and to measure their affinity values.

We apologise for the omission. We now clearly state in the revised version that this experiment has been performed by using FluPolA containing PB1 577E and PA 556R (legend to Fig. 1e). In fact, we have tested the polymerase with PB1 577K and PA 556Q, obtaining similar results (see Figure 1 below). We decided against including these data in the manuscript, as all other experiments were performed using FluPolA with PB1 577E and PA 556R (we have now also included a statement in the Introduction that all experiments, including structural analysis and functional assays, have been performed using FluPolA containing PB1 K577E and PA Q556R) and these data in our opinion would not add any important information to the manuscript. However, we would be happy to include these data as a supplementary figure if advised to do so.

Figure 1. Pull-down showing wildtype Tky05 FluPolA, with PB1 577K instead of E, and PA 556Q instead of R, binding to RNAP II S5P CTD peptide (yellow arrow), but not in the presence of ANP32B (black arrow). ANP32B is present in threefold molar excess over FluPolA, shown in input lanes. Controls include serine 2 phosphorylated (S2P) CTD peptide, unphosphorylated (unphos) peptide, and scrambled peptide. Details of the peptides are described in Martínez-Alonso et al J Virol 2016.

Some specific Points:

1. Fig. 1b: The side chain of K635 in Fig. 1b requires improvement for a clearer representation.

We improved the visibility of the K635 in Fig. 1b by repositioning the label and changing the angle of the picture for this panel.

2. On Page 5, Line 121, and Page 6, Line 130, the "product exit channel" and "template exit channel" are mentioned. Labeling these channels in the corresponding figures (Fig. 1b and Fig. 1d) would enhance reader comprehension.

Illustrating these channels in Fig. 1b and 1d result in these figures becoming overly complicated and difficult to interpret. Therefore, we opted for generating new supplementary

figure panels (see Supplementary Fig. 1c) to illustrate these channels; we now refer to these figure panels at the appropriate lines in the Results section.

3. In Figure 2c, it is recommended to indicate the interface from FluPol_E or FluPol_R for better contextualization.

There is no FluPol_E and FluPol_R interface in this figure panel. This panel shows a close-up view of the PA^{Endo}, PB2^{Lid} and PB2^{CBD} interface in FluPol_E.

4. Page 6, Line 127. The expression "while in the present conformation..." is ambiguous. Providing PDB numbers and clarifying the statement would enhance clarity.

We have re-phrased the statement and included PDB numbers in the legends to Supplementary Fig. 1b.

5. Please fix the bond angle outliers in the PDB validation report.

We thank the Reviewer for pointing this out; we are working on this re-opening the PDB deposition and correcting the single outlier.

Reviewers' Comments:

Reviewer #1:

Remarks to the Author:

The authors have provided thorough responses to the review comments and have addressed all my concerns and suggestions. The manuscript has improved, and I recommend publication of this work.

Reviewer #2:

Remarks to the Author:

I think the revised manuscript is much clearer and more coherent and the author has addressed the major concerns. Based on my assessment of the revised manuscript, I don't have more comments to the manuscript.